# Multimodal Early Birth Weight Prediction Using Multiple Kernel Learning

**DOI:** 10.3390/s24010002

**Published:** 2023-12-19

**Authors:** Lisbeth Camargo-Marín, Mario Guzmán-Huerta, Omar Piña-Ramirez, Jorge Perez-Gonzalez

**Affiliations:** 1Departamento de Medicina Traslacional, Instituto Nacional de Perinatología Isidro Espinosa de los Reyes, Montes Urales 800, Lomas de Virreyes, Miguel Hidalgo, Mexico City 11000, Mexico; lisbethcamargo@yahoo.com.mx (L.C.-M.); mguzmanhuerta@yahoo.com.mx (M.G.-H.); 2Departamento de Bioinformática y Análisis Estadístico, Instituto Nacional de Perinatología Isidro Espinosa de los Reyes, Montes Urales 800, Lomas de Virreyes, Miguel Hidalgo, Mexico City 11000, Mexico; delozath@gmail.com; 3Unidad Académica del Instituto de Investigaciones en Matemáticas Aplicadas y en Sistemas, Universidad Nacional Autónoma de México, Km 4.5 Carretera Mérida-Tetiz, Municipio de Ucú, Yucatán 97357, Mexico

**Keywords:** multimodal learning, fetal medicine, ensemble feature selection, multimodal data

## Abstract

In this work, a novel multimodal learning approach for early prediction of birth weight is presented. Fetal weight is one of the most relevant indicators in the assessment of fetal health status. The aim is to predict early birth weight using multimodal maternal–fetal variables from the first trimester of gestation (Anthropometric data, as well as metrics obtained from Fetal Biometry, Doppler and Maternal Ultrasound). The proposed methodology starts with the optimal selection of a subset of multimodal features using an ensemble-based approach of feature selectors. Subsequently, the selected variables feed the nonparametric Multiple Kernel Learning regression algorithm. At this stage, a set of kernels is selected and weighted to maximize performance in birth weight prediction. The proposed methodology is validated and compared with other computational learning algorithms reported in the state of the art. The obtained results (absolute error of 234 g) suggest that the proposed methodology can be useful as a tool for the early evaluation and monitoring of fetal health status through indicators such as birth weight.

## 1. Introduction

Growth is one of the main indicators of fetal health, its importance lies in the fact that it can translate the previous maternal nutritional status, the epigenetic factors involved in its developmental environment and the family genetic potential. The percentile value of growth during all stages of pregnancy allows for classifying deviations from it (intrauterine growth restriction and macrosomia), establishing a monitoring method and defining the behaviors that best determine the stage and mode of birth [1]. Currently, the clinically validated hypothesis on fetal programming establishes that fetal weight is a determinant in neonatal morbidity, the onset of metabolic disorders in childhood (obesity) and the presence of pathologies in adulthood, such as diabetes, metabolic syndrome, hypertension, cardiovascular disease, kidney disease, among others [2]. According to the World Health Organization, in 2019, a total of 2.4 million newborn deaths were reported, with premature births, infections, birth complications and congenital defects being the main causes [3].

In this context, the early prediction of Birth Weight (BW) can be an important indicator for evaluating fetal health and facilitating timely interventions before birth. In clinical practice, formulas such as Hadlock, Campbell or Shepard are traditionally used for fetal weight estimation, which are fed with indicators derived from Fetal Biometry by Ultrasound. However, these equations show high error rates (up to 25%) and only provide an estimation of fetal weight at the time of the fetometry [4,5]. Therefore, it is necessary to propose new strategies for the accurate and early estimation of fetal BW.

The aim of this work is to predict early birth weight using multimodal maternal–fetal variables from the first trimester of gestation. For this purpose, a new methodology based on the Multiple Kernel Learning algorithm fed with a subset of optimally selected biomarkers is proposed. Early prediction can help specialists monitor and evaluate fetal health status.

### State of the Art

Several researchers have recently implemented machine learning algorithms for fetal birth weight classification. Desiani et al. [6] proposed a Naive Bayes classifier to discriminate according to the weight condition of infants (LBW: Low Birth Weight, NBW: Normal Birth Weight and HBW: High Birth Weight). For this, the classifier’s inputs were attributes, such as maternal age, maternal gravida, maternal education level, birth-to-birth interval and hypertension conditions. Similar work was reported by Faruk et al. [7], who implemented the Random Forest (RF) and Binary Logistic Regression algorithms to classify between a population with LBW and non-LBW. As variables, they incorporated the place of residence, mother and father’s education, as well as the mother’s age and occupation. Moreira et al. [8] implemented classifiers based on Decision Trees and Support Vector Machines (SVM), which incorporated clinical variables, hypertension and obesity data and symptoms. The target was to classify fetuses with LBW. Jezewski et al. [9], implemented a neural network that utilizes descriptors derived from cardiotocographic monitoring (which consists of three signals: fetal heart rate, uterine contractions and fetal movements). The purpose was to predict fetuses with LBW.

Other authors have focused their work on fetal weight estimation using computational or deep learning algorithms. Feng et al. [10] performed fetal weight estimation using the Deep Belief Network algorithm, whose input features were metrics derived from Fetal Biometry by US acquired before birth, and some maternal measurements such as abdominal circumference and maternal fundal height. Khan et al. [11] collected data from various hospitals to estimate fetal birth weight. The proposed methodology includes a feature selection stage using algorithms such as Correlation-based Feature Selection, Classifier Attribute Evaluator or Principal Components Analysis (PCA); additionally, they implemented several regression algorithms, such as Support Vector Regression (SVR), RF, Linear Regression, Sequential Minimal Optimization Regression or Multilayer Perceptron (MLP). These algorithms were fed with variables obtained before delivery, among which anthropometric data of the mother, mother’s education and systolic and diastolic blood pressure in the second and third trimesters of gestation stand out. Alzubaidi et al. [12] proposed a methodology for segmentation and extraction of indicators from US images of the fetal head using neural networks. They trained several regression models (RF, Linear Regression, SVR, LASSO, Voting Regressor or Deep NN among the main ones), using metrics such as Biparietal Diameter (BPD), Head Circumference (HC) and Occipitofrontal Diameter (OFD) to estimate the fetal weight at the time of US acquisition. Tao et al. [13] proposed a Hybrid Long Short-Term Memory Network that takes into account temporal metrics derived from fetometry and maternal weight during the third trimester of gestation for predicting BW. Lu et al. [14] used an ensemble model consisting of RF, XGBoost and Light Gradient Boosting Machine algorithms to estimate fetal weight. They proposed curve fitting and the integration of maternal anthropometric variables to feed the regression algorithm. The curve fitting was performed for indicators derived from fetometry. Plotka et al. [15,16] proposed algorithms based on neural networks with transforms to estimate fetal weight hours before birth. The proposed algorithms are fed with fetal ultrasound images and/or the combination of images and tabular data obtained mainly from fetal biometry. Other authors have proposed approaches based on SVR for fetal weight estimation. Sereno et al. [17] proposed an SVR based on indicators obtained from fetometry and the umbilical artery resistance index. Yu et al. [18,19] implemented the Fuzzy-SVR and Evolutionary Fuzzy-SVR algorithms trained with indicators derived from fetometry by US.

The described works have shown good performances for fetal weight classification or estimation. However, those investigations present the following disadvantages: they do not conduct a rigorous analysis of feature contribution; most of the works are based on descriptors obtained close to birth (which is not considered an early prediction of the BW); no study has been conducted considering data modality; and, to our knowledge, no reported work has performed birth weight prediction using first-trimester gestational variables.

The main contribution of this work is the early prediction of fetal birth weight using maternal–fetal variables from the first trimester of gestation, which, to our knowledge, has not been published before. To achieve this, we propose the implementation of the supervised learning algorithm Multiple Kernel Learning (MKL), which is based on SVMs and a set of optimal kernels that help map each analyzed data modality to a new space. As an additional contribution, we propose an ensemble strategy of feature selection algorithms for four data modalities to be analyzed: Fetal Biometry by US, Maternal Anthropometric data, Doppler, and Maternal US. It is worth mentioning that there is a published proceeding with preliminary results [20]. In that work, an SVR was implemented; however, no feature selection strategy was proposed, and the dataset used was incomplete.

## 2. Materials and Methods

The proposed pipeline is presented in Figure 1. The methodology begins with the organization of the four modalities of maternal–fetal data used: Anthropometrics, Fetal US Biometry, Doppler and Maternal US. Next, an exploratory data analysis was carried out by means of a correlation map. This is followed by an analysis of the contribution of each feature through the combination of a set of feature selection algorithms. For birth weight estimation, the Multiple Kernel Learning algorithm is implemented, which optimally selects the best kernel for each data modality. Finally, the validation stage is performed by comparing actual birth weight vs. estimated weight. The following subsections detail each stage.

### 2.1. Data Description and Analysis

The data were acquired by obstetrics experts from the National Institute of Perinatology of Mexico, “Isidro Espinosa de los Reyes”. This database contains records of 578 participants; for each record, a set of 18 maternal–fetal variables from four different modalities were obtained: Anthropometric data, Fetal Biometry, Doppler and Maternal US. The variables were obtained in the first trimester of pregnancy (between weeks 9 and 14 of gestation, according to crown–rump length). Afterward, the weight was measured moments after birth. Table 1 presents the details of the obtained variables. All participants provided their informed consent in accordance with the Helsinki Declaration for the use of information in the present study. To observe the relationships between all the data, a first analysis using a Pearson correlation map was carried out.

### 2.2. Ensemble Feature Selection

Recently, various strategies have been proposed for ensemble techniques of feature selection algorithms [21,22,23]. The ensemble involves combining two or more feature selection approaches and subsequently feeding them into a machine learning algorithm. This approach has been employed in several medical applications such as breast cancer diagnosis [24], Parkinson’s disease [25], in datasets of COVID-19, lung cancer, malaria and hepatitis, among others [26]. Those studies demonstrated that the ensemble of feature selection algorithms can help prevent overfitting, enhance computational efficiency, as well as improve performance in classification or regression.

For this work, an ensemble of several feature selection algorithms is proposed with the aim of enhancing performance in BW estimation, as well as identifying the most important features in the regression process. The implemented algorithms are:**Normalized Mutual Information (NMI):** is a measure used to assess the similarity between two sets of data, considering the joint and individual information of each set [27]. For this work, the NMI metric is utilized to measure the descriptive capability of each feature versus the BW in a univariate approach.**F-Statistic:** is a statistical measure used to determine the significance of a regression model. It assesses whether at least one of the independent variables in the model has a significant impact on the dependent variable [28].**Least Absolute Shrinkage and Selection Operator (LASSO):** is a linear model with restrictions that allows for variable selection, given that after an iterative selection of the alpha parameter, the weights associated with non-relevant variables become exactly zero [29].**Multiple Linear Regression (LR):** is an algorithm that fits a linear model to minimize the quadratic error between multiple independent variables and the outcome variable [30]. Upon the assumption that all input variables are scaled to the same intervals, the absolute values of the weights derived from the LR model may be interpreted as indicators of the relative importance of each variable.**Mean Decrease in Impurity (MDI):** is a feature selection algorithm, based on an ensemble of decision trees. This algorithm quantifies the importance of individual variables within the regression model process, employing mean squared error as the criterion for impurity assessment [31]. In the computation of MDI, two variants were contemplated: Random Forest (MDI-RF) and Extra Trees (MDI-ET), each comprising 500 estimators and utilizing the conventional impurity criterion.

The algorithms under consideration are grouped based on their characteristics into: univariate statistics, encompassing NMI and F-Statistic; multivariate linear regression, including LASSO and LR; and decision trees-based, namely MDI-RF and MDI-ET. In the final stage of feature selection, the outcomes of these algorithms were scaled between 0 and 1. Given *d* as the total number of features analyzed, the selected set of *Q* features, denoted as EFS, was computed as follows:(1)$$\begin{array}{ccc} A_{d} & = & \frac{1}{N}\sum_{i=1}^{N}FS_{i}^{\left(d\right)}, \end{array}$$(2)$$\begin{array}{ccc} EFS & = & \max_{Q}\left\{A_{1},A_{2},\;\cdots ,\;A_{d}\right\}, \end{array}$$
where $FS_{i}$ denotes the feature selection outcome of the *i*-th algorithm, $A_{d}$ is the feature selection outcome average for the *d*-th feature and *N* indicates the total number of algorithms implemented. To identify the most significant features, the EFS method incorporates the *Q*-highest $A_{d}$ scores, which in combination minimizes the prediction error.

### 2.3. Multiple Kernel Learning

Multiple Kernel Learning (MKL) is a supervised learning algorithm proposed by Sonnenburg et al. [32]. It is a widely used approach in various medical applications due to its high performance and ability to integrate information from diverse modalities such as neuroimages [33], genomic data [34] or physiological signals [35], for example.

Given a set of *N* training samples, ${\left\{(x_{i},y_{i})\right\}}_{i=1}^{N}$, where $x_{i}\in R^{d}$ is an input vector, *d* represents the dimension or number of features across all modalities, and $y\in R^{+}$ are the output values. The aim is to accurately predict an output $\hat{y}$ for a new given input *x*, by a regression function based on MKL [32,36]. This function can be denoted as:(3)$$\hat{y}=\sum_{i=i}^{N}\left(\alpha_{i}^{*}-\alpha_{i}\right)K\eta \left(x,x_{i}\right)+b,$$
where the coefficients α are the Lagrange multipliers obtained during training, *b* is the bias term, $K\eta (x,x_{i})$ is the combination of kernel functions applied to a new input *x* and training samples $x_{i}$. Finally, kernel combination was performed as a weighted linear sum, which is expressed as:(4)$$K\eta \left(x,x_{i}\right)=\sum_{m=1}^{P}\eta_{m}k_{m}\left(x^{m},x_{i}^{m}\right),$$
where $\eta_{m}$ denotes the kernel weights for each of the *m* data modalities. For this work, the kernel functions used were linear, polynomial and/or radial basis [37]. For the combination of kernels, the best kernel was selected for each maternal–fetal data modality (as can be seen in Figure 1). To obtain the optimal weights $\eta_{m}$, as well as to evaluate the best kernel functions $k_{m}$ and their corresponding parameters, a grid search optimization strategy was used during the cross-validation. The target of this stage is to accurately predict BW using the MKL algorithm trained with the previously selected multimodal variables from the EFS stage.

### 2.4. Validation

For validation, from the total dataset (N = 578), 500 records were randomly selected for a five-fold Cross-Validation (CV), and the remaining 78 were used for a final unseen-data test (hold-out set). The performance metrics used to assess the accurate prediction of BW were the Mean Absolute Error (MAE) and the Mean Absolute Percentage Error (MAPE), denoted as:(5)$$MAE=\frac{1}{S}\sum_{i=1}^{S}\left|y_{i}-{\hat{y}}_{i}\right|,$$
(6)$$MAPE=\frac{100\%}{S}\sum_{i=1}^{S}\left|\frac{y_{i}-{\hat{y}}_{i}}{y_{i}}\right|,$$
where *S* is the sample size, $y_{i}$ represents the real values and ${\hat{y}}_{i}$ represents the estimated BWs. For this study, the MAE metric measures the absolute error expressed in grams, and MAPE measures the error expressed in terms of percentage.

In the preliminary works [20,38], the SVR algorithms with different kernels and Gaussian Processes were implemented, finding better performances for SVR. Therefore, in order to compare the results obtained from the proposed approach, the Random Forest Regressor, an SVR and an Artificial Neural Network were implemented. These algorithms, as well as their optimized hyperparameters using the grid search approach, are described below:**Random Forest Regressor (RF):** is a supervised algorithm based on the ensemble of decision trees [39]. For its implementation, the number of estimators, the criterion (mean squared error, mean absolute error, Friedman squared error or Poisson criterion) and the tree’s depth were optimized.**Support Vector Regressor (SVR):** is a generalization of a Support Vector Machine for continuous variable prediction [36]. Due to the preliminary results obtained [20], a radial basis kernel was used for this algorithm, and the γ coefficient and regularization parameter *C* were optimized.**Artificial Neuronal Network (ANN):** is an algorithm based on multi-layer perceptrons [40]. For this model, the activation function (identity, logistic, tanh or ReLU), the solver (LBFGS, SGD or Adam), the L2 regularization term, batch size, learning rate and the number of neurons in the hidden layers were optimized.

All algorithms for BW estimation were evaluated with the features from each data modality individually (Anthropometric data, Fetal Biometry, Doppler and Maternal US), using all features, and with the subset of features previously selected by the EFS algorithm. The algorithms, validations and graphs were implemented in MATLAB 2023a, Python 3.11, scikit-learn 1.3.2 and the MKL framework [41].

## 3. Results and Discussion

In this section, the obtained results and their discussion regarding the correlation analysis, feature selection and the performances presented in BW prediction are shown. The results begin with the correlation map shown in Figure 2. In this figure, positive correlations are represented in blue, and negative correlations in red. Overall, it can be observed that no descriptor has a high correlation with respect to birth fetal weight. Among the highest correlations are MW and MBMI with 0.9, the PT/PL ratio and PT with −0.7, as well as various correlations among the indicators derived from UAPI (blue cluster with values ranging from 0.6 to 0.9). These results demonstrate that individually, no maternal–fetal variable has a linear relationship with BW.

The results of the feature selection analysis using EFS are presented in Figure 3. In this figure, the result of each feature selection algorithm individually, as well as their collective integration can be observed. The highest cumulative contribution is 1, and 0 corresponds to the lowest importance. The final subset of selected features included MH, MW, PL, MBMI, DBP, CRL, L-UAPI and FL. It is worth noting that this subset includes descriptors from each of the four data modalities studied. On the other hand, it can be noted that the features with the lowest contribution are SEX, R-UAPI, AC, PT, as well as MIN-, MAX- and MEAN-UAPI. The poor contribution of the UAPI-based descriptors may be due to the high correlation that exists among them (see Figure 2), with L-UAPI having the highest contribution according to the analysis conducted.

The performances during cross-validation using data from each modality independently can be observed in Table 2. Overall, it can be seen that the ANN algorithm achieved the worst performances in estimating BW, with errors ranging from 305 to 398 g. In contrast, the MKL, RF and SVR algorithms presented similar performances, with SVR being the algorithm that achieved the best results, with average errors ranging from 263 to 332 g. When comparing the performances achieved by each data modality, it can be observed that the results corresponding to Fetal Biometry show the best scores, with average errors ranging from 262 to 347 g, followed by Anthropometric variables (MAE from 262 to 305 g), Maternal US (MAE from 264 to 327 g) and Doppler (MAE from 332 to 398 g). In the global evaluation, the best result achieved was in the combination of data derived from Fetal Biometry and the MKL algorithm, with an MAE of 262.4 ± 19 g; however, it can be observed that the performances by the MKL and SVR algorithms are similar. It could be due to the fact that, at this stage, there is no integration of multimodal data, and therefore, only one kernel is optimized in the MKL approach, which is similar to the SVR algorithm.

The results of the final test with unseen data from each modality are shown in Table 3. In general, the errors achieved are higher than those presented during cross-validation (Table 2). In this case, the best result obtained in predicting BW is once again for the combination of variables derived from Fetal Biometry and the MKL algorithm, with an MAE of 283 g. However, the SVR-based approach consistently shows the best overall performances for each data modality. Similar to what was shown during cross-validation, algorithms fed with Fetal Biometry variables present the lowest average errors (ranging from 283.7 to 361.5 g), followed by Anthropometric data (MAEs between 285 and 314 g), Maternal US (MAEs from 292 to 350 g) and Doppler (MAEs from 350 to 402 g).

In Table 4, the results of cross-validation using all maternal–fetal features and a subset of descriptors selected by the EFS algorithm are shown. It can be observed that the results achieved with the subset of previously selected features are better than the approach using all variables together (Table 4, columns 2 and 3) and individually by each modality (Table 2). Furthermore, it can be noted that the MKL algorithm and the subset of features selected by EFS are the combination that presents the best performance (MAE of 233.4 g and MAPE of 8.48%) compared to all the unimodal and multimodal analyses conducted.

Finally, in Table 5, the results of the final test with unseen data using all variables and automatic feature selection are presented. It can be noticed that the selected subset of variables helps improve the performance of all four regression algorithms. Another aspect to highlight is that the performances improve when combining the four modalities compared to the results obtained from individual modalities (Table 2 and Table 3). Furthermore, consistent with the trend shown during cross-validation (Table 4), the combination of the MKL regressor and the EFS algorithm presents the best performance with an MAE of 234.7 g and a MAPE of 8.32%. The results found suggest that the approach based on selecting the optimal kernel (MKL algorithm) for each data modality, in combination with appropriate feature selection for each modality, contributes to a better estimation of BW.

When comparing the results obtained with those reported in the state of the art, it was found that Feng et al. [10] report estimates of BW in a range from 179 to 245 g. However, the variables used in the prediction were acquired prior to birth. Khan et al. [11] report MAEs ranging from 294 to 352 g; in their study, the authors collected maternal or fetal data from various hospitals primarily during the second and third trimesters of pregnancy. In [14], they report a 7% percentage error in BW estimation using a curve-fitting approach based on anthropometric variables. On the other hand, Yu et al. [19] report percentage errors between 6.6% and 7.5%, using the Evolutionary Fuzzy SVR algorithm. In the previous work [20], preliminary results were reported using an SVR with different kernels and a smaller sample set. The obtained results in that study show an absolute error ranging from 287 to 304 g. In our study, the best results achieved during the final test were an MAE of 234.7 g and a percentage error of 8.32%, which is competitive or better than what is reported in the state of the art. It is important to highlight that, unlike the proposed methodology, previous works are based on maternal or fetal features obtained during the second and third trimesters of pregnancy or prior to birth, and do not conduct an analysis of the data by modality [10,11,12,13,14,17,18,19].

Unlike previously published research, this study exclusively utilizes variables obtained in the first trimester of gestation for predicting BW through a multimodal regression approach. The obtained results suggest that this methodology can be useful as an early fetal weight prediction tool, which can assist obstetric specialists in conducting an early assessment of fetal health.

## 4. Conclusions

In this study, a novel approach to supervised multikernel learning is presented. The system was fed by an optimized subset of maternal–fetal variables acquired in the first trimester of pregnancy. The challenge is to predict fetal birth weight early on.

The methodology begins with an exploratory data analysis using a correlation map. In this analysis, no linear relationships correlated with birth weight were found. Subsequently, a feature selection algorithm based on an ensemble of six algorithms with different properties (univariate approaches, linear fitting-based and decision tree-based) was proposed. As a result of this analysis, the most important variables derived from the four data modalities studied (Anthropometric data, Fetal Biometry, Doppler and Maternal US) were selected. To estimate BW, the optimized MKL algorithm was implemented with the best kernel for each data modality. The weighted multi-kernel approach helps map each analyzed data modality to an optimal new space with the aim of enhancing performance during regression. The results were validated and compared with other widely used algorithms in the state of the art. The performances observed in predicting BW are comparable to or better than other reported methodologies, which use descriptors obtained before birth or during the second and third trimesters of pregnancy. In contrast to what has been published in the state of the art, the proposed early BW prediction approach is fed only with variables from the first trimester of gestation.

Weight is one of the main indicators of fetal health; a large number of functional, structural, chromosomal, genetic and infectious alterations have a direct impact on fetal weight. However, the complexity of obtaining the exact fetal weight during pregnancy is a clinical problem that many have tried to solve for years. Intrauterine growth restriction is defined as the inability of the fetus to reach its “growth potential”. In obstetrics, fetal weight is the closest indicator to know the “growth potential”; therefore, it is crucial to be able to estimate adequate and early indicators such as birth weight.

In future work, we expect to validate this system with new multicenter clinical studies; also, we intend to incorporate algorithms for automatic measurement of fetal US indicators such as DBP, HC, FL and AC, among others. Another point that we would like to include for future work is the evaluation of different algorithms for the optimization of the different parameters and the selection of kernel functions.

In conclusion, this work presents a new birth weight prediction system based on a nonparametric learning approach fed with multimodal maternal and fetal ultrasonographic variables measured in the first trimester of pregnancy. The results suggest that the proposed methodology can be a useful tool, which will provide a better follow-up of growth during pregnancy. 

## Figures and Tables

**Figure 1 sensors-24-00002-f001:**
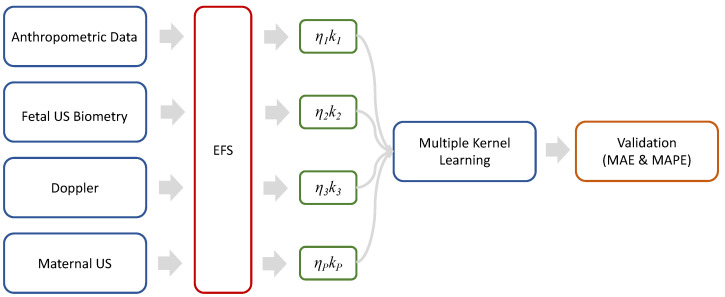
Representative diagram of the proposed methodology. On the left-hand side, the four types of analyzed maternal–fetal data are shown, followed by the Ensemble Feature Selection (EFS) approach (this algorithm selects the best variables for each data modality). Subsequently, the different kernels $k_{m}$ and their weighting factors $\eta_{m}$ are shown for each data modality m=1,2,⋯,P. The Multiple Kernel Learning regressor is trained to predict the BW. Mean Absolute Error (MAE) and Mean Absolute Percentage Error (MAPE) metrics are proposed for validation.

**Figure 2 sensors-24-00002-f002:**
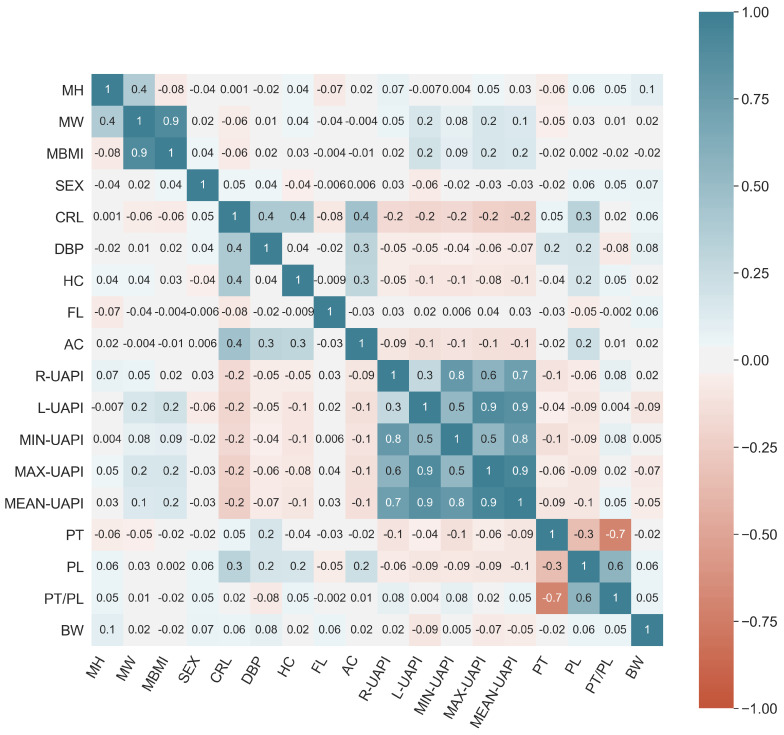
Correlation matrix of the various maternal–fetal variables used. Red values indicate negative correlations, and blue values represent positive relationships.

**Figure 3 sensors-24-00002-f003:**
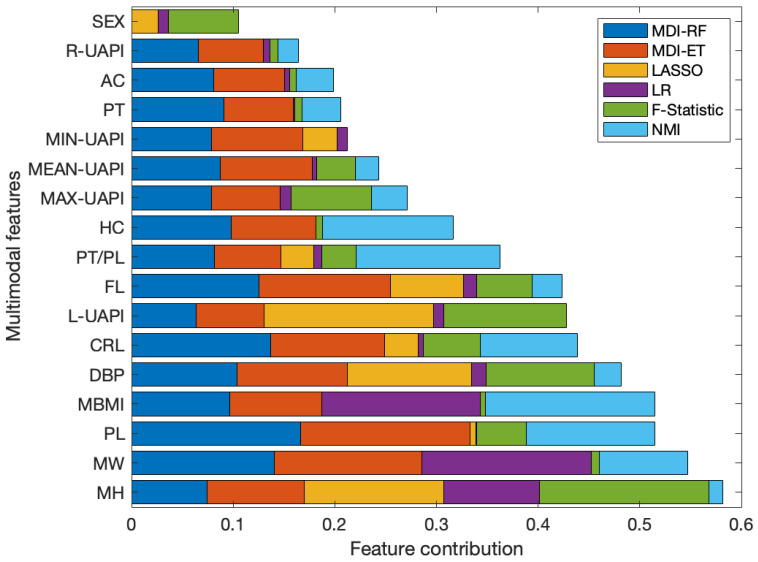
Results of the Ensemble Feature Selection sorted according to their cumulative contribution. Each color represents the contribution normalized according to the proposed feature selection algorithms.

**Table 1 sensors-24-00002-t001:** Description of the maternal–fetal features used (mean ± standard deviation).

Feature	Statistical Description
**Anthropometric data**	
Maternal Height (MH)	155.60 ± 5.96 cm
Maternal Weight (MW)	63.67 ± 11.05 kg
Maternal Body Mass Index (MBMI)	26.28 ± 4.22 kg/m^2^
Sex	284/294 (F/M)
**Fetal US Biometry**	
Crown–Rump Length (CRL)	66.72 ± 9.91 mm
Biparietal Diameter (BPD)	21.34 ± 5.90 mm
Head Circumference (HC)	75.10 ± 16.62 mm
Femur Length (FL)	13.27 ± 16.86 mm
Abdominal Circumference (AC)	64.78 ± 12.16 mm
**Doppler**	
Right Uterine Artery Pulsatility Index (R-UAPI)	1.51 ± 0.70
Left Uterine Artery Pulsatility Index (L-UAPI)	1.61 ± 0.92
Minimum Uterine Artery Pulsatility Index (MIN-UAPI)	1.26 ± 0.52
Maximum Uterine Artery Pulsatility Index (MAX-UAPI)	1.86 ± 0.95
Mean Uterine Artery Pulsatility Index (MEAN-UAPI)	1.56 ± 0.65
**Maternal US**	
Placental Thickness (PT)	2.05 ± 1.09
Placental Length (PL)	7.37 ± 1.72
Placental Thickness/Length Ratio (PT/PL)	4.15 ± 1.48
**Target**	
Birth Weight (BW)	2868.95 ± 333.68 g

**Table 2 sensors-24-00002-t002:** Results of 5-fold cross-validation (mean ± standard deviation) with data from each modality. The lowest errors in BW prediction are shown in bold.

Regressor	Anthropometric Data		Fetal US Biometry		Doppler		Maternal US	
	MAE (g)	MAPE (%)	MAE (g)	MAPE (%)	MAE (g)	MAPE (%)	MAE (g)	MAPE (%)
MKL	264.1 ± 23	9.81 ± 0.9	**262.4 ± 19**	**9.54 ± 0.9**	338.7 ± 20	11.26 ± 0.9	264.5 ± 21	9.81 ± 0.8
RF	**262.4 ± 20**	**9.57 ± 0.9**	265.7 ± 16	9.66 ± 0.8	340.1 ± 23	11.32 ± 0.9	270.4 ± 24	9.83 ± 0.8
SVR	263.1 ± 22	9.78 ± 0.9	264.1 ± 21	9.80 ± 0.9	**332.2 ± 21**	**11.06 ± 0.9**	**264.1 ± 22**	**9.80 ± 0.9**
ANN	305.1 ± 24	11.14 ± 1.3	347.3 ± 25	12.43 ± 1.0	398.2 ± 23	13.27 ± 1.3	327.5 ± 25	11.71 ± 1.0

**Table 3 sensors-24-00002-t003:** Results of the final test with unseen data from each modality (hold-out set). The lowest errors in BW prediction are shown in bold.

Regressor	Anthropometric Data		Feta US Biometry		Doppler		Maternal US	
	MAE (g)	MAPE (%)	MAE (g)	MAPE (%)	MAE (g)	MAPE (%)	MAE (g)	MAPE (%)
MKL	286.5	10.32	**283.7**	**10.25**	352.7	12.19	292.4	10.40
RF	297.1	10.60	295.9	12.11	367.7	12.72	**292.1**	**10.39**
SVR	**285.4**	**10.31**	285.3	10.31	**350.4**	**12.13**	293.7	10.45
ANN	314.7	11.42	361.5	12.63	402.4	13.92	350.9	12.39

**Table 4 sensors-24-00002-t004:** Five-fold cross-validation performances (mean ± standard deviation) using all descriptors and feature selection by the ESF algorithm. The lowest errors in BW prediction are shown in bold.

Regressor	All Features		Automatic Feature Selection by EFS	
	MAE (g)	MAPE (%)	MAE (g)	MAPE (%)
MKL	**255.3 ± 20**	**9.31 ± 0.4**	**233.4 ± 18**	**8.48 ± 0.7**
RF	264.7 ± 21	9.65 ± 0.8	256.4 ± 21	9.54 ± 0.9
SVR	261.3 ± 22	9.52 ± 1.1	242.9 ± 19	8.84 ± 0.8
ANN	262.8 ± 20	9.58 ± 0.9	258.4 ± 20	9.39 ± 0.7

**Table 5 sensors-24-00002-t005:** Results of the final test using all descriptors and a subset of features selected by the EFS algorithm (all samples belong to the unseen dataset). The lowest errors in BW prediction are shown in bold.

Regressor	All Features		Automatic Feature Selection by ESF	
	MAE (g)	MAPE (%)	MAE (g)	MAPE (%)
MKL	**265.4**	**9.59**	**234.7**	**8.32**
RF	276.7	9.99	245.7	8.73
SVR	272.3	9.84	250.2	8.89
ANN	280.1	10.11	242.3	8.62

## Data Availability

The data used in this research can be shared via email.

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
