# Peer review of "Multimodal Early Birth Weight Prediction Using Multiple Kernel Learning"

_sensors, 2023, doi:10.3390/s24010002_

Round 1

Reviewer 1 Report

Comments and Suggestions for Authors

The problem of early and accurate prediction of birth weight is the subject of a number of scientific studies, with a view to timely assessment of possible difficulties both during birth and in relation to the likelihood of future health problems in childhood and adulthood. The contribution of the authors, to the difference from the published data of other similar studies, consists in the possibility for early prediction of the birth weight on the basis of multimodal maternal-fetal parameters from the first trimester of gestation. The authors have a previous similar publication, but in this one they further develop the support vector regression algorithm by applying an ensemble strategy from feature selection from 4 data modalities to be analysed: Fetal Biometry by ultrasound, Maternal Anthropometric data, Doppler, and Maternal ultrasound. The applied methodology for features selection, data analysis and procedures for birth weight prediction is correct. The obtained results of average error 234.7g and a percentage error of 8.32% are comparable with reported in the state-of-the art.

The abstract correctly presents the content of the article. The presented review and analysis in part “Introduction” demonstrate expertise and deep knowledge of the authors in the field. Reference sources are relevant to the content and are cited at appropriate places in the text.

Remarks and questions:

Just a small editorial note. Ln 83 "most of the work ..." should be "most of the works ...”.

Author Response

Dear Reviewer,

In the attached document you will find the response to the comments and in the new manuscript you will find the modifications in blue color.

Best regards

Reviewer 2 Report

Comments and Suggestions for Authors

This article presents a novel approach for early birth weight prediction using a wide range of maternal-fetal data gathered during the first trimester of pregnancy. This dataset includes anthropometric measurements and metrics from fetal biometry, Doppler, and maternal ultrasound. The methodology begins with feature selection using an ensemble-based approach, followed by the application of a nonparametric Multiple Kernel Learning regression algorithm. However, this article provides a useful Ensemble Feature Selection method that combines different kernels to improve accuracy, its level of innovation is relatively modest. Furthermore, there are some suggestions may help authors to polish your paper.

1. The representation of contributions is not clear in the section of introduction.

2. In Figure 1, the definition of symbols is not same with Figure.

3. The description of methods is too simple to show the concept and process of your proposed methods in mathematical way.

4.  In the experimental setup, the parameter configuration appears to lack a comprehensive exploration of optimal performance, as it relies on a single fixed value.

5. The experiment section of the article is somewhat constrained by the relatively small number of comparative experiments conducted.

Comments on the Quality of English Language

need to double check language

Author Response

(The authors gave the same response as above.)

Reviewer 3 Report

Comments and Suggestions for Authors

Dear Authors,

- I did not see any most recently published related work to fetal birth weight estimation. I found several papers regarding to the fetal birth weight estimation, which should be included in your study as follow:

1. Płotka, S., Grzeszczyk, M. K., Brawura-Biskupski-Samaha, R., Gutaj, P., Lipa, M., Trzciński, T., & Sitek, A. (2022, September). BabyNet: residual transformer module for birth weight prediction on fetal ultrasound video. In International Conference on Medical Image Computing and Computer-Assisted Intervention (pp. 350-359). Cham: Springer Nature Switzerland.
2. 
Płotka, S., Grzeszczyk, M. K., Brawura-Biskupski-Samaha, R., Gutaj, P., Lipa, M., Trzciński, T., ... & Sitek, A. (2023). BabyNet++: Fetal birth weight prediction using biometry multimodal data acquired less than 24 h before delivery. Computers in Biology and Medicine, 107602.

Please consider it to your discussion, to compare with image, or image+tabular data and their results.

- please fit your tables to the template

However, the study sound technically good, and shows important task which can be applied to the worldwide community.

Author Response

(The authors gave the same response as above.)

Round 2

Reviewer 2 Report

Comments and Suggestions for Authors

Authors answered the majority of comments

Comments on the Quality of English Language

The representation of English should be improved.

There are still many tiny grammar errors in the manuscript, authors would better double check.